# Using Low-Cost Measurement Systems to Investigate Air Quality: A Case Study in Palapye, Botswana

**William Lassman [1,*,†], Jeffrey R. Pierce [1], Evelyn J. Bangs [1], Amy P. Sullivan [1], Bonne Ford [1], Gizaw Mengistu Tsidu [2], James P. Sherman [3], Jeffrey L. Collett, Jr. [1] and Solomon Bililign [4]**

[1]  Department of Atmospheric Science, Colorado State University, 1301 Campus Delivery, Fort Collins, CO 80523, USA; jeffrey.pierce@colostate.edu (J.R.P.); ejb.bangs@beyondbb.com (E.J.B.); Amy.Sullivan@colostate.edu (A.P.S.); bonne@atmos.colostate.edu (B.F.); Jeffrey.Collett@colostate.edu (J.L.C.J.)

[2]  Department of Earth and Environmental Sciences, Botswana International University of Science and Technology, Private Bag 16, Palapye, Botswana; mengistug@biust.ac.bw

[3]  Department of Physics and Astronomy, Appalachian State University, 525 Rivers Street, Boone, NC 28608, USA; shermanjp@appstate.edu

[4]  Department of Physics, North Carolina A&T University, 306 Marteena, 302 Gibbs, Greensboro, NC 27411, USA; bililign@ncat.edu

*  Correspondence: lassman1@llnl.gov

†  Now at, Lawrence Livermore National Laboratory, Atmosphere, Earth and Energy Division, 7000 East Ave., Livermore, CA 94550, USA.

**Abstract:** Exposure to particulate air pollution is a major cause of mortality and morbidity worldwide. In developing countries, the combustion of solid fuels is widely used as a source of energy, and this process can produce exposure to harmful levels of particulate matter with diameters smaller than 2.5 microns ($PM_{2.5}$). However, as countries develop, solid fuel may be replaced by centralized coal combustion, and vehicles burning diesel and gasoline may become common, changing the concentration and composition of $PM_{2.5}$, which ultimately changes the population health effects. Therefore, there is a continuous need for in-situ monitoring of air pollution in developing nations, both to estimate human exposure and to monitor changes in air quality. In this study, we present measurements from a 5-week field experiment in Palapye, Botswana. We used a low-cost, highly portable instrument package to measure surface-based aerosol optical depth (AOD), real-time surface $PM_{2.5}$ concentrations using a third-party optical sensor, and time-integrated $PM_{2.5}$ concentration and composition by collecting $PM_{2.5}$ onto Teflon filters. Furthermore, we employed other low-cost measurements of real-time black carbon and time-integrated ammonia to help interpret the observed $PM_{2.5}$ composition and concentration information during the field experiment. We found that the average $PM_{2.5}$ concentration (9.5 µg·m$^{-3}$) was below the World Health Organization (WHO) annual limit, and this concentration closely agrees with estimates from the Global Burden of Disease (GBD) report estimates for this region. Sulfate aerosol and carbonaceous aerosol, likely from coal combustion and biomass burning, respectively, were the main contributors to $PM_{2.5}$ by mass (33% and 27% of total $PM_{2.5}$ mass, respectively). While these observed concentrations were on average below WHO guidelines, we found that the measurement site experienced higher concentrations of aerosol during first half our measurement period (14.5 µg·m$^{-3}$), which is classified as "moderately unhealthy" according to the WHO standard.

**Keywords:** $PM_{2.5}$; low-cost measurements; Botswana; air quality; air pollution

## 1. Introduction

Exposure to particulate air pollution with aerodynamic diameters smaller than 2.5 μm ($PM_{2.5}$) is linked with premature mortality [1] and morbidity [2], and is currently one of the most important causes of premature mortality in developing and rapidly industrializing nations [3]. State-of-the-art methods for quantifying the global health burden of air pollution use remote sensing tools and atmospheric chemical transport models (CTMs) fused with in situ measurements to produce estimates of pollutant concentrations in regions where surface measurements are sparse [4,5]. There is uncertainty associated with each step in this process that propagate through health-impact assessments [6,7]. Therefore, despite advancements in CTM and remote sensing capabilities, there is a need for in situ measurements to develop process-based understanding of the sources and chemical processes that drive local and regional scale population exposure to $PM_{2.5}$. Specifically, estimates of mortality due to air pollution exposure in sub-Saharan Africa are large [7], and in situ measurements of air pollution are rare.

Over 40% of the population in sub-saharan African nations uses domestic biomass burning for heating and cooking [3,8]. However, estimates of emissions from domestic biomass burning vary [9,10], leading to uncertainty in the representation of these sources in regional and global CTMs, which propagates to estimates of disease burden [7]. Moreover, as sub-saharan African nations undergo economic development, the important regional sources of aerosols will change, as coal combustion for electricity production replaces domestic solid fuel use, and fossil fuel vehicles become more common [11]. Therefore, many different measurement studies of various air pollutants and precursors will become increasingly important to capture these changes in emissions and to reduce uncertainties of population-level exposure. Furthermore, "sub-saharan Africa" refers to a large collection of countries with different climate regions and cultures, spanning an area three times the size of the United States. While there have been some measurements of combustion emissions in some locations in southern Africa, combustion emissions vary considerably from one country to the next, and the number and duration of studies currently in the literature is inadequate to accurately represent the entire region.

Botswana is a medium-sized African nation. While sparsely populated, it is experiencing population growth and rapid economic development and industrialization [12]. Consequently, the sources, and therefore the magnitude, composition, exposure, and impact of human exposure to $PM_{2.5}$ is rapidly changing. According to recent global estimates, air pollution in Botswana is not currently a major public health concern [3], but may become one as economic development progresses [11]. Additionally, biomass burning is widely used in Botswana for cooking and heating, and the use of household solid fuels has been directly linked to poor health outcomes in the capital city of Gaborone [13]. However, Botswana also produces electricity using coal power plants; according to official estimates, domestic electrical generating capacity has increased by over 50% in the last three years [12]. Coal power plants emit precursors for sulfate and sulfuric acid, and there are multiple examples of coal power plant emissions impacting ambient $PM_{2.5}$ concentrations, which in turn affect human health [1]. Furthermore, increases in the size of the Botswana automobile fleet, which has more than doubled in size since 2002 [12], can also contribute to $PM_{2.5}$ exposure. Because of these rapid changes, the role $PM_{2.5}$ plays in public health in Botswana is also likely changing, though due to a lack of available measurements, it is difficult to assess.

In situ air quality monitoring traditionally has required expensive instrumentation and technical expertise to operate the instruments continuously. The cost of conducting these measurements can be prohibitive for many developing nations, and thus datasets to evaluate satellite/model estimates of air pollution exposure, such as from the Global Burden of Disease (GBD) [3], are often unavailable. Therefore, there is a need to develop lower-cost approaches for assessing air pollution. The development of low-cost sensors to measure atmospheric pollutants such as $PM_{2.5}$, ozone, $NO_X$ [14] is an active area of research. Many of these sensors sacrifice accuracy and precision in order to reduce manufacturing and operational costs; therefore, they may not be suitable for answering some research questions or in situ monitoring applications [15]. However, lower cost-per-sensor may allow for the deployment of more sensors, and integration with Internet of Things (IoT) technology may allow for distributed sensor

networks, which can open the door for new research methodologies [16]. It is important to understand the technical limitations of these instrument platforms, and how the tradeoff in accuracy and precision can impact the types of research and monitoring applications that are feasible. Furthermore, there are other practical limitations of the use of new sensor platforms in real-world scenarios that are difficult to anticipate without pilot-scale studies. These sensors must be evaluated over a range of expected scenarios, using established methods or data sources, and other technical challenges associated with the deployment and successful operation of these low-cost sensors must be conducted in realistic field-measurement scenarios.

In June 2018, a collaboration between North Carolina A&T University (NCAT), Appalachian State University (APP), and Botswana International University of Science and Technology (BIUST) was started to establish a long-term record of surface-based sunphotometer measurements of aerosol optical depth (AOD); the atmospheric column-integrated aerosol light extinction) at 500 nm and 670 nm, using custom-built sunphotometers developed and tested at APP NASA Aerosol Robotic Network (AERONET) site. During a 5-week visit to establish the measurement protocol, we also deployed a suite of low-cost instruments in order to measure $PM_{2.5}$ surface concentration and composition in this region during the initial 5-week measurement period. The primary instrument that we deployed during the 5 weeks was the Aerosol Mass and Optical Depth (AMOD) device, developed for a Citizen Science project (CEAMS) [17,18]; the AMOD collects $PM_{2.5}$ mass on a Teflon filter, while simultaneously conducting a surface-based aerosol optical depth measurement and measuring real-time $PM_1$, $PM_{2.5}$, and $PM_{10}$ from a third party optical sensor (Plantower PMS 5003). In addition to the AMOD, we collected time-integrated samples of gas-phase ammonia, as well as real-time black carbon concentrations, using two low-cost commercial measurements. To our knowledge, these are the first measurements of aerosol concentrations and composition in this region in Botswana that have been published. In this study, we present the results of the measurements and post-measurement analysis. Here we will compare the total measured $PM_{2.5}$ concentrations to GBD estimates, as well as present speciated-PM results, put the PM measurements in context of AOD measured by the AMOD and satellites, and briefly discuss the important sources for $PM_{2.5}$ in this region.

## 2. Methods

### 2.1. Site Description and Important Aerosol Sources

The measurements were made on the campus of the Botswana International University of Science and Technology (BIUST) located in Palapye, Botswana. Palapye is a village located at 22.59° S, 27.12° E, more than 150 km from the nearest major metropolitan area (Figure 1) within a semi-arid climate zone characterized by hot and dry weather, with less than 300 mm of precipitation annually. The measurements were made between 19 June and 20 July 2018, corresponding to southern hemisphere winter. Daytime temperatures generally ranged between 25 and 30 °C, with nighttime lows between 0 and 6 °C. There was little or no cloud cover during our measurement period. However, from 5–10 July, the weather was overcast and cool with some light precipitation, associated with a synoptic-scale system [19]. The area surrounding Palapye consists of arid shrubland with sandy soils. Blowing and suspended dust from arid soils is an important source of coarse-mode particulate matter in this region [20], though traditionally dust is not a major source of $PM_{2.5}$. Livestock is a major part of Botswana's economy [20], and this land is frequently used for cattle grazing as well. Cattle and other livestock can be a source of ammonia, which is a precursor for $PM_{2.5}$ [21–23].

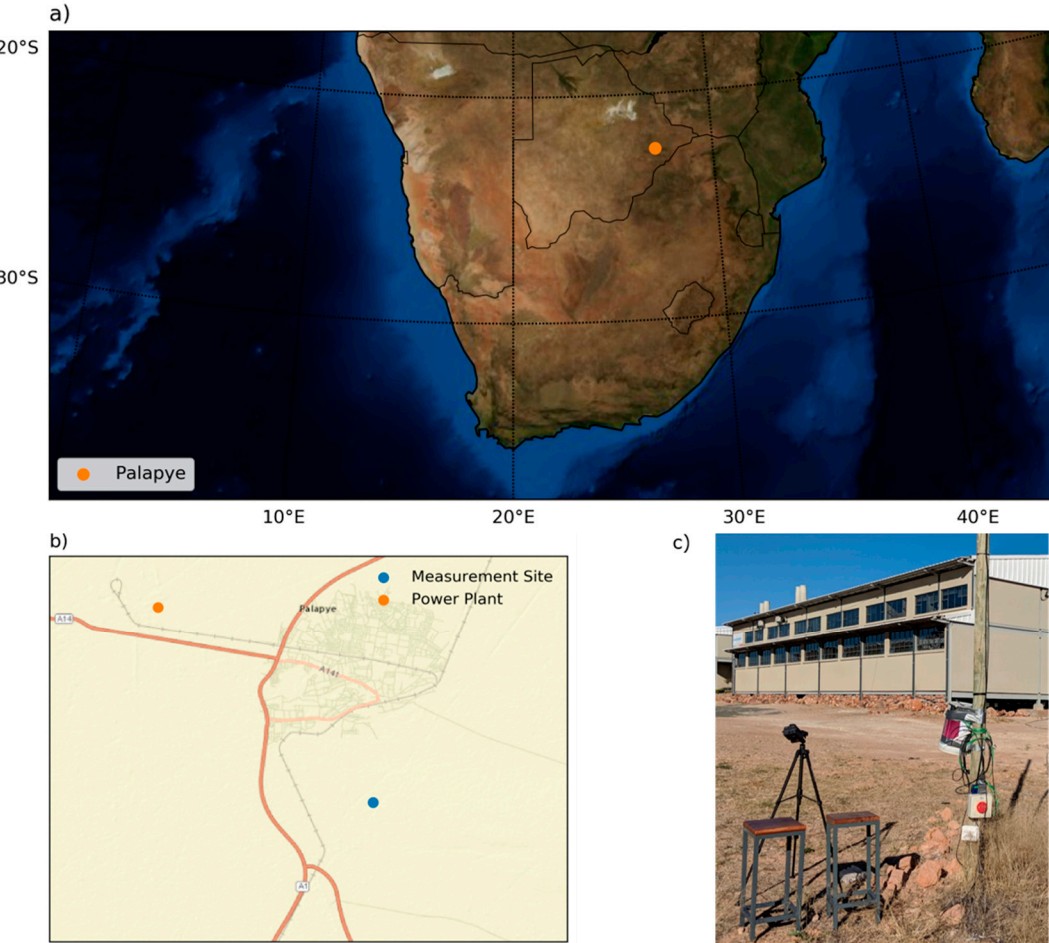

**Figure 1.** (**a**) Location of Palapye within Botswana. (**b**) Map of Botswana International University of Science and Technology (BIUST) campus relative to the village of Palapye, the A1 Highway, and the Morupule power station (mean winds are east-north-easterly). (**c**) Photograph of the experimental setup with the CEAMS Aerosol Mass and Optical Depth (AMOD) on the black tripod, the Radiello Passive Ammonia sampler under the bucket and the AethLabs MicroAethalometer$^{TM}$ mounted to the wooden utility pole. Map generated using World Street Maps (credit: Esri, HERE, Garmin, USGS, Intermap, INCREMENT P, NRCan, Esri Japan, METI, Esri China-Hong Kong, NOSTRA, ©OpenStreetMap contributors, and the GIS user community). ©OpenStreetMap contributors 2019. Distributed under a Creative Commons BY-SA License.

In addition to these regional sources of aerosol, there are important anthropogenic sources as well. Traditionally, developing countries rely on domestic solid-fuel combustion for energy, which is an important source of carbonaceous aerosol and is a major health concern [3]. In Botswana, studies have shown that domestic biomass burning can impact ambient air quality in the capital city Gaborone [24] and has a detectable effect on pediatric pneumonia health outcomes [13]. While the rest of Botswana also uses solid-fuel combustion for energy [25], no measurement studies have been performed outside of Gaborone. In addition to domestic biomass burning, Botswana has some industrial sources, such as metal processing and coal combustion [20]. A notable example near Palapye is the Morupule Power Station, a coal power plant with an adjacent coal mine to provide fuel for the power station. The Morupule Power Station has been operational since 1990 but has undergone a recent expansion to a nominal capacity of 600 MW, although it was not operating at 100% capacity during our sampling period. Although it is a functioning power plant, it is not listed as operational in emission inventories used in the most-recent modelling study of $PM_{2.5}$ for this region [11], so modelling studies will not capture its contributions to ambient air quality. Coal power plants are known sources of $SO_2$,

which reacts in the atmosphere to form sulfate and sulfuric acid, and $NO_X$, which can affect ozone formation and form nitrate and nitric acid; while certain emission control technologies may be effective at reducing the emissions [26], it is unknown whether such controls are operational on this power station. However, as the capacity of this power station increases and/or control technologies are changed, the impact on regional air quality will continue to evolve.

In addition to the power plant, Palapye experiences a variety of other potential sources of $PM_{2.5}$. The most recent census estimates indicate that the population of Palapye was 36,000 in 2011. However, the village is undergoing rapid population growth and the 2018 population is thought to be closer to 80,000. In addition to the establishment of BIUST (target size 6000 students, plus employees), Palapye is the site of rapid construction and development, including construction of a new commercial district to support the growing population. Moreover, the growing automobile fleet, both locally and nationally, are increasing the potential impact of traffic emissions on our measurement site. The A1, which is the busiest road in Botswana, travels directly through the town center and within 5 km of our measurement site. Vehicles in Botswana commonly use diesel fuel, which can emit more black carbon and organic aerosol precursors than other gasoline vehicles [27].

Figure 1b denotes the location of our sampling site relative to the BIUST campus, the villages of Khurumela and Palapye, the Morupule Power Station, and the A1. Outside of the region shown on the map, the surrounding area is sparsely populated desert shrubland with light animal grazing. In Figure 1c, we show our experimental setup with the CEAMS AMOD instrument mounted on a tripod, and the MicroAethelomter, and ammonia measurement secured to the wooden utility pole approximately 1.3 m off the ground. The measurements are approximately 50 m from the nearest structure, as shown in Figure 1c.

## 2.2. Measurements

To perform many of the measurements, we used the Aerosol Mass and Optical Depth (AMOD) measurement system [18], developed with Access Sensor Technologies. The AMOD AOD measurement uses 4 photodiodes at 440, 520, 680, and 870 nm to measure optical extinction along these wavelengths. The AMOD also uses the same cyclone-filter system to collect $PM_{2.5}$ on Teflon filters from the more-widely used Ultrasonic Personal Air Sampler (UPAS) sensor [28] to collect samples of $PM_{2.5}$ for laboratory characterization described below. Finally, the AMOD devices include a third-party Plantower PMS5003 sensor, which estimates $PM_{10}$, $PM_{2.5}$, and $PM_1$ concentrations; however, these size bin values are based on a theoretical model rather than measurements [29]. The Plantower PMS5003 has a resolution of $1~\mu g \cdot m^{-3}$, and the instrument specifications suggest that it can effectively measure particles with a diameter greater than 0.5 μm (98% efficiency, only 50% efficiency for 0.3 μm), with concentrations in the range of $0–500~\mu g \cdot m^{-3}$ [30]. However, evaluation studies have suggested that it underpredicts particles in the small size bins and overpredicts in the large size bins [29]. Thus, we only show $PM_{2.5}$ measurements here, which previous studies have shown to have good correlation with reference monitor methods [18,29,31,32]. The AMOD measurement system has been extensively tested in low-aerosol conditions in Colorado, and validation of the measurement system can be found in [18], and an example of its application for local air quality studies can be found in [17].

In this study, we used two AMOD devices to perform the measurements, deploying the sensors alternately while allowing the other sensor to recharge. Each sensor was deployed for 48 h, with the exception of the final two measurements which were 24-h deployments. Each filter collected $PM_{2.5}$ for the duration of the measurement period, using a $2~L~min^{-1}$ flow rate of ambient air. We collected 14 total samples, with an additional 6 filters that were transported to the measurement site for use as blanks. The Teflon filters were sealed inside plastic containers before and after deployment. Filters were weighed pre and post deployment, and were further characterized by Sootscan$^{TM}$ Black Carbon technique and X-ray fluorescence (XRF) to detect inorganic elements. Additionally, we extracted each filter in 15 mL DI water with sonication without heat for 40 min. The aqueous extracts were analyzed for anions and cations using ion-exchange chromatography on a Thermo Fisher Dionex ICS-3000.

Anions were measured using AS-14A column employing a sodium carbonate/sodium bicarbonate eluent at 1 mL/min. Cations were determined using an AS12A column using 20 mM methanesulfonic acid eluent at 0.5 mL/min. WSOC (water-soluble organic carbon) was also determined for each extract using a Sievers M9 Turbo Total Organic Carbon (TOC) Analyzer. The analyzer converts the carbon in the sample to carbon dioxide using chemical oxidation and UV light. The carbon dioxide formed is measured by conductivity. Because the analyzer only measures the mass of carbon, we estimate the total mass of water-soluble carbonaceous aerosol (WSOC mass) by multiplying the mass of carbon detected by 1.6, which is a typical ratio of total mass to carbon mass for fresh combustion emissions [33]. We used the average concentration of the WSOC and ions determined from the blank filters to blank-correct our measurements. The Limit of Detection (LOD) for WSOC was 17.1 µg per filter and 10.0 µg per filter for the aggregate IC mass; the minimum measured values for WSOC and IC were 14.5 and 6.3 µg per filter, and average values were 21.6 and 24.1 µg per filter, respectively.

In addition to the AMOD, we used a microAeth[(R)] AE51 aethalometer (Aethlabs, San Francisco, CA, USA) to measure real-time black carbon concentrations. We used a 150 mL/min flow rate, and a sampling timebase of 5 min. The AE51 battery life was approximately 18 h, then it would need 12 h to recharge before being redeployed. The instrument was deployed at variable times during the day, and retrieved the following morning, where it was allowed to charge to full battery before redeployment. Therefore, the exact time of day of instrument deployment varied.

To measure the ambient gas-phase $NH_3$, we used Radiello[TM] passive $NH_3$ adsorption cartridges inside diffusive bodies. The cartridges were deployed for 3–4 days, depending on the cartridge, and stored in an air-tight plastic centrifuge vial for transport to/from Botswana. Each live sampling cartridge was transported with a corresponding carrying blank to quantify contamination due to transport and storage. The cartridges were extracted in 6 mL DI water in 16 mL Falcon test tubes using sonication without heat. The $NH_4^+$ (ammonium) ionic concentration was determined in the aqueous extract using the same cation chromatography method described above. The LOD was 3.6 ppb.

### 2.3. Ancillary and Remote Observations, and Back-Trajectory Modelling

In addition to the measurements that we conducted, we made use of other available data. BIUST maintains a meteorological instrument station that provides measurements of wind speed, wind direction, temperature, relative humidity, and solar flux at hourly resolution. For satellite observations, we use the Moderate Resolution Imaging Spectroradiometer (MODIS) instrument aboard the Terra and Aqua polar-orbiting satellites. The MODIS instrument consists of a 36 radiometer wavelengths; the AOD product is available twice daily, in mid-morning and early afternoon (~10:30 a.m. and 1:30 p.m. local time). In this analysis, we use the Level 2 Land and Ocean retrieval product at 10 km resolution [34]. The Land and Ocean primarily uses the Deep Blue retrieval but uses Dark Target for pixels where Dark Target is thought to perform better.

Finally, we use NOAA HYSPLIT back-trajectory modelling to trace air parcels backwards to determine what sources may have impacted these air masses. For the HYSPLIT trajectories, we used the Frequency Back trajectory method. We initiated trajectories every 6 h, starting at 500 m above the surface, for the duration of the field measurement period. We aggregated the trajectory frequencies for the first and second halves of the measurement period (15–30 June, and 1–15 July, respectively).

## 3. Results and Discussion

### 3.1. Concentrations at BIUST

First, we present the $PM_{2.5}$ time series from the AMOD devices in Figure 2: the real-time Plantower observations are denoted by the points, distinguishing between the two AMOD instruments; the filter observations are denoting by the horizontal black lines with the width representing the duration of the measurement period (all measurements were for 48 h except for the last two, which were 24 h duration). As shown in the timeseries, the Plantower sensor on AMOD 2 was offline after the third

deployment of that sensor. Additionally, the sensors were not deployed during the overcast/raining period from 5 July to 10 July.

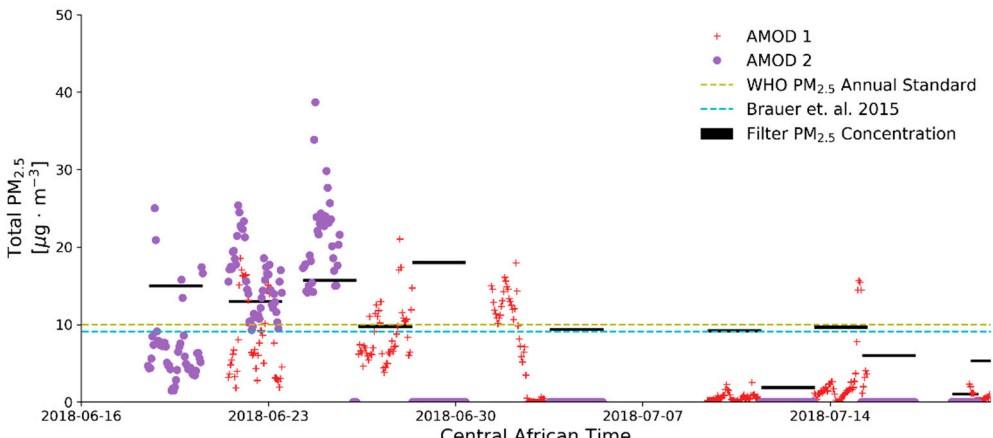

**Figure 2.** Time series plot of real-time PM$_{2.5}$ and filter PM$_{2.5}$ from the two AMOD instruments, with the estimates from the Global Burden of Disease (GBD) Report [3] and the World Health Organization annual standard overlaid.

In Figure 2, the filter and Plantower measurements both show that the PM$_{2.5}$ concentrations before (18 June–4 July) are generally greater than the concentrations after the overcast period (10–20 July). The average concentration during the first half of the measurement campaign was 14.0 µg·m$^{-3}$, as compared to 6.5 µg·m$^{-3}$ during the second half of the measurement period, according to the filter-based measurements. In Figure 2, we have overlaid the World Health Organization (WHO) [35] standard for annual average PM$_{2.5}$ concentrations (10 µg·m$^{-3}$) and the estimates for the 10 km by 10 km box around our measurement site in the 2015 GBD report (9.1 µg·m$^{-3}$). The actual average concentration during this time period was 9.4 µg·m$^{-3}$, which is very close to the GBD estimate and below the WHO annual limit [4]. However, given the limited duration of our measurements and high variability in the concentrations we observed, it is difficult to conclusively determine the actual mean concentration, or whether the "high-concentration period" or "low-concentration period" is more common in Palapye, or how these may vary seasonally. Therefore, we will attempt to use other information to identify the mechanism for this variability, in order to extract some more generalizable information about air quality in Palapye. In addition to our measurements of mean PM$_{2.5}$ concentration, we also have the characterization of the filters that we collected to provide some information on the PM$_{2.5}$ composition. Additionally, we can use the Plantower and MicroAeth observations to learn about the daily variability in PM$_{2.5}$ and black carbon, an important tracer for combustion. We will discuss the composition information in Section 3.2 and the temporal variability in Section 3.3.

*3.2. Filter PM$_{2.5}$ Composition Characterization*

In Figure 3, we show the breakdown of the filter PM$_{2.5}$ samples by different compositional types: inorganic ionic species, water-soluble organic carbon species (WSOC), and black carbon (BC). We plot the residual PM$_{2.5}$ mass in blue and refer to it as "unspeciated" PM$_{2.5}$. Each filter is blank corrected and plotted where the bar width is reflective of the sample duration (as in Figure 2). Overlapping bars indicated time periods where the two AMOD instruments were simultaneously collecting samples. From Figure 3, we see that: (i) the most important speciated fraction of the PM$_{2.5}$ mass is inorganic in nature, (ii) the black carbon mass is typical for PM$_{2.5}$ with some combustion influence, but (iii) there is little WSOC (often less than BC) on these filter samples and (iv) there remains a significant unspeciated fraction. Aerosol types that we are unable to directly measure that could constitute the unspeciated fraction are insoluble organic aerosol species or dust. However, we do not expect dust to be an important fraction of this unspeciated aerosol mass, both based on the X-ray diffraction results

that will be discussed shortly. The presence of BC suggests that combustion from either domestic solid fuel use or diesel vehicles is impacting the $PM_{2.5}$ in this region. Given that Palapye is too remote to be significantly influenced by long-range transport of urban emissions in general, it is likely that the combustion is generally local. As organic aerosol undergoes oxidative aging in the atmosphere, the aerosol O/C ratio increases [36]; this typically increases the solubility of the organic aerosol species in water. Because the combustion emissions impacting our measurement site may be local, the chemical compounds may have undergone less oxidation and are therefore less soluble in water.

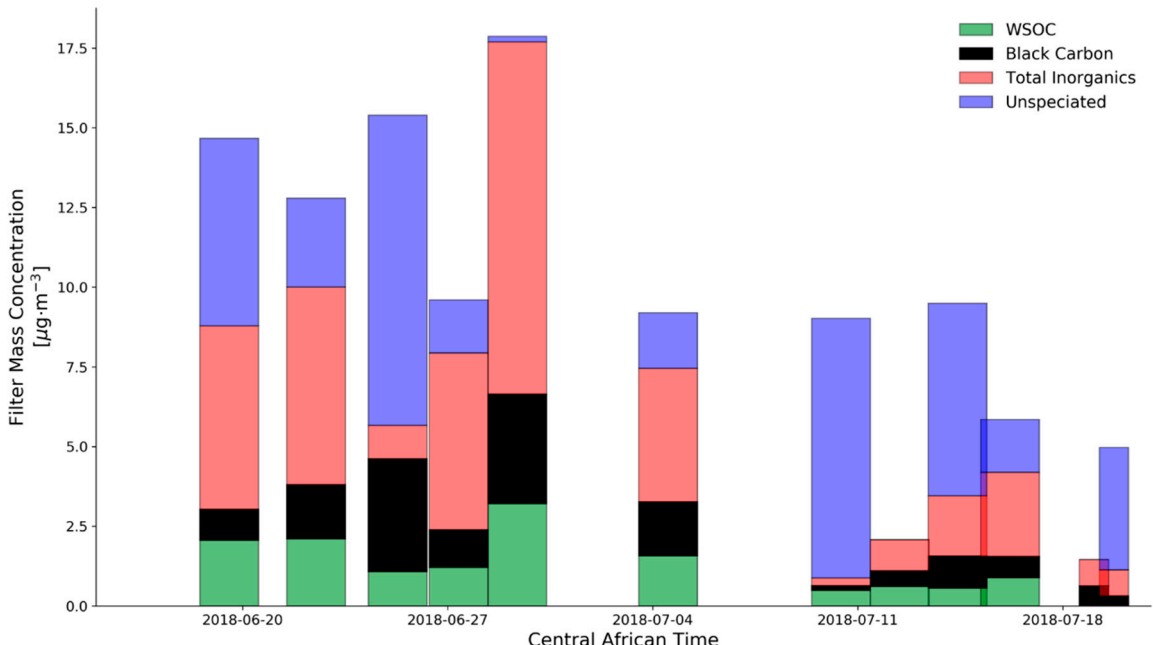

**Figure 3.** Results of analytical characterization of filters, showing the mass concentration of black carbon (black, water-soluble organic aerosol mass (green), inorganic aerosol mass (red), and the residual unspeciated aerosol mass (blue). The width of each bar is the duration of the filter sample; overlapping bars represent filters with overlapping sampling periods.

The most important single constituent of the aerosol by mass is the inorganic ionic fraction of the aerosol. The mass concentrations for the common ionic aerosol species from the ion chromatography characterization are shown in Figure 4. By mass, sulfate is the most abundant ionic species in the particle phase, followed by ammonium. Sulfate and sulfuric acid is most commonly formed from the oxidation of $SO_2$ by two oxidation pathways: gas-phase oxidation by OH radical, and aqueous-phase oxidation inside cloud droplets. These two processes have very different timescales associated with them; the gas-phase mechanism has a lifetime of approximately 1 week, while the aqueous-phase mechanism is much faster. Given the lack of clouds in the region during the first two weeks, the gas-phase mechanism is likely dominant for this measurement site during this time period. While the Moropule Power Plant is located within 10 km of the measurement site, and is likely a large regional source of $SO_2$, it is unlikely that the oxidation of $SO_2$ is occurring quickly enough to produce lots of sulfate at the measurement site. To produce the observed mass of sulfate so close to the source of $SO_2$, either the air in the region must have stagnated to prevent ventilation of $SO_2$ emissions, allowing the production of sulfate over a period of a few days. While there is evidence that some stagnation occurred during the first half of the measurement period, the prevailing winds never directed emissions from the powerplant to the measurement site. However, another possibility is that the measured sulfate is regional background. South Africa generates much more electricity annually than Botswana, including electricity that is exported to Botswana and other surrounding nations. Due to South Africa's possession of rich coal deposits, coal-fire power plants are ubiquitous in the north east of the country. Previous modeling

studies have shown that the prevailing wind during summer is easterly and northeasterly over Botswana thereby allowing transport of biomass burning aerosols from northeastern parts of South Africa [37,38]. In [38], biomass burning aerosol from northeastern and central parts of South Africa is shown to be an important regional source of aerosol during summer. However, as compared to other events, such as various anthropogenic activities in South Africa, open biomass burning activities emit lower amounts of $SO_2$ during this time of the year [39]. Therefore, it is not obvious that advection of the regional sulfate from the east and northeast of South Africa could account for a substantial fraction of the observed sulfate in Palapye.

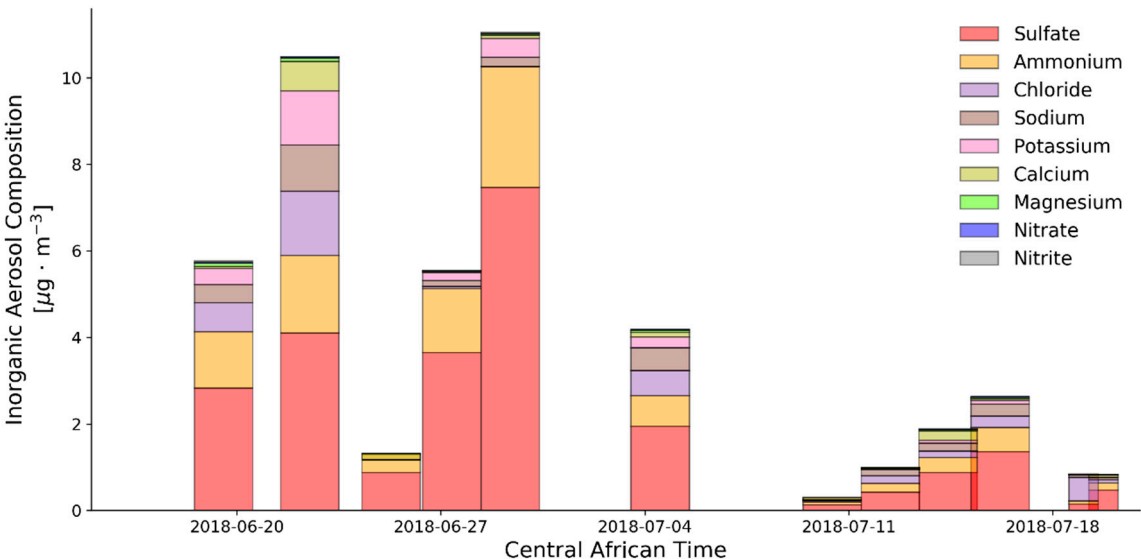

**Figure 4.** Ion chromatography aerosol composition results.

The second-most important inorganic species in the aerosol phase is $NH_4^+$, followed by sodium, chloride, potassium, and other salts that are usually associated with sea salt aerosol or dust. The masses of $NH_4^+$ and $SO_4^{2-}$ on each day in Figure 4 generally correspond to a molar ratio of 2:1, meaning $SO_4^{2-}$ is completely or nearly completely neutralized by $NH_4^+$. In addition to the AMOD filter analysis, we conducted time-integrated measurements of gas-phase $NH_3$ at 3.5-day resolution. However, none of the measurements indicated any gas-phase $NH_3$ above the limit of detection of 3.6 ppb. It seems unlikely that there was enough total ammonia to nearly completely neutralize $SO_4^{2-}$ each day, yet gas phase $NH_3$ was never above the 3.6 ppb detection limit. We have identified two potential artifacts of the filter analysis that may be a reason for $SO_4^{2-}$ but $NH_3$ below its detection limit. (i) The AMOD filter system has no secondary filter or denuder system to account for evaporation of semi-volatile species such as ammonium nitrate or semi-volatile organic aerosol, that may have previously deposited on the filter in the particle phase [40]; we see no evidence of nitrate in our filters, but this may be due to off-gassing of ammonium nitrate during or after the measurement period. Given that sulfate was fully neutralized, it is possible that ammonium nitrate formed if enough nitric acid was present. If ammonium nitrate was present and was ammonia limited (e.g., excess nitric acid relative to ammonia), ammonia equilibrium vapor pressures could have been orders of magnitude lower than the 3.6 ppb ammonia detection limit as verified by E-AIM (http://www.aim.env.uea.ac.uk/aim/aim.php; [41]). The presence of ammonia-limited ammonium nitrate could explain the neutralized sulfate but ammonia below the detection limit throughout the measurement period; however, we do not have nitric acid measurements in the region to verify. (ii) Ammonium may actually have been too low to neutralize sulfate, which would cause ammonia concentrations to be below detection limits, but the filter was exposed to ammonia at some point between the measurement period and analysis (potentially due to offgassing of ammonia from the 3D-printed AMOD chassis). It is possible that total ammonia concentrations

in the region are in fact low; while livestock is an important part of the Botswana economy [42], and livestock production is a known source of $NH_3$ to the environment, the free-range manner in which Botswana cattle are raised are not conducive to the massive ammonia emissions observed near more industrialized cattle operations in the United States [22], Australia [43], Canada [44,45] and Europe [46,47]. In short, due to the low-cost nature of the measurement system, there are many potential sources of error in the measurement of various inorganic semi-volatile aerosol constituents. However, we expect the quantification of sulfate and other non-volatile species (e.g., sodium, chloride, potassium) to be more robust.

In addition to the IC analysis, we conducted x-ray fluorescence (XRF) analysis of the filters to identify the metallic species present in the aerosol; the results of the XRF analysis are shown in Figure 5. Many of the atomic species detectable by the XRF analysis are also present in ions detected by IC. If the IC ions dominate the form of the XRF atom (e.g., low amounts of organic sulfur), the XRF and IC analysis may agree. For example, the mass of sulfur atoms obtained via XRF agrees well with the mass of ionic sulfate, as determined via the IC. Likewise, there is agreement for Cl and Na species from the two methods. However, the XRF also allows us to see the quantity of silicon and iron in the aerosol, which are tracers for dust in this region. Based on the results in Figure 5, the dust in this region is not important for $PM_{2.5}$ particles. However, this does not preclude that suspended dust is present in coarse particles, and dust is known to be important in this region for climate radiative effects and visibility [20,48].

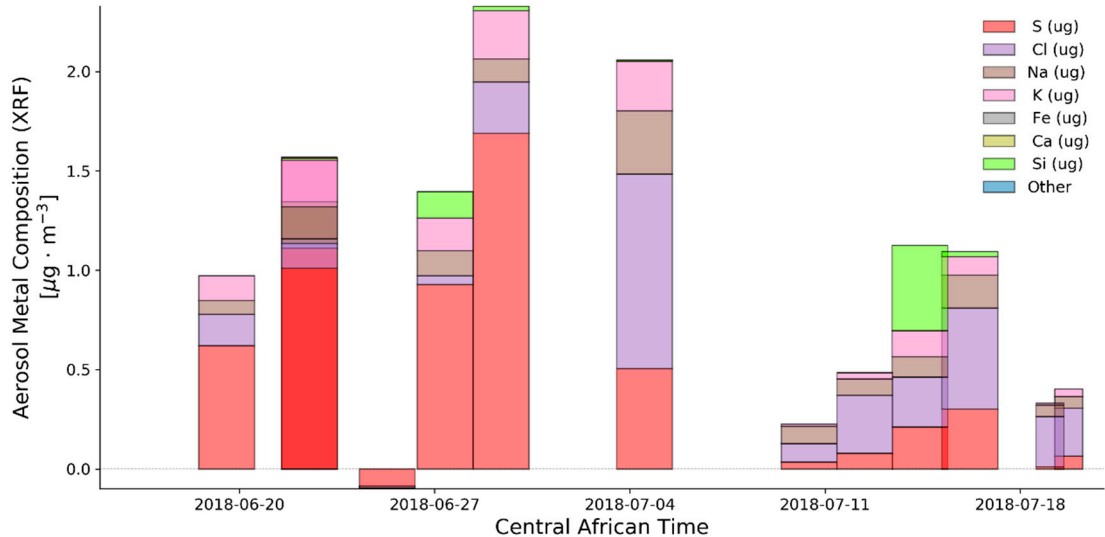

**Figure 5.** X-Ray Fluorescence analysis results from filter samples; colors for important species correspond to related ions in IC results in Figure 4. Values below the zero line indicate that there was no mass on the filter above the detection limit.

### 3.3. Temporal Variability

In addition to the filter samples, we use the Plantower real-time $PM_{2.5}$ measurement, along with the MicroAeth black carbon measurements to investigate the diurnal variability in the aerosol at the BIUST measurement site. We plot the Plantower data for the entire time period in Figure 6a with the black carbon concentrations using a different scale on the second Y axis. In Figure 6a, we see that the concentrations for total $PM_{2.5}$ were much higher during the first half of the time period, and comparatively lower during the second half, which is corroborated by the filter-based measurements as described above. There appears to be a difference in black carbon concentrations, but it is much less pronounced than the difference in total $PM_{2.5}$. We will revisit this difference between the first and second halves of the measurement period shortly.

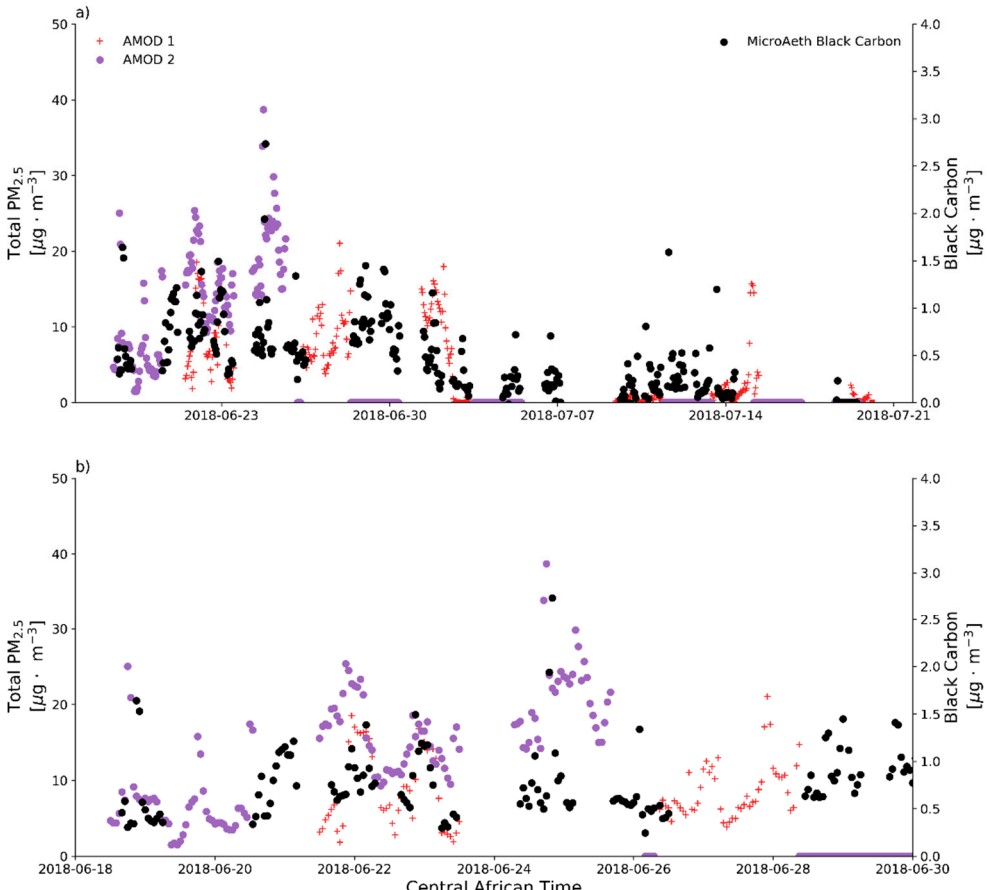

**Figure 6.** Time series of plantower PM$_{2.5}$ and black carbon (**a**) for the entire time period and (**b**) zoomed in on the first half of the time period, where a regular diurnal cycle is present.

In Figure 6b, we show only the first half of the measurement period, when concentrations were elevated. The total PM$_{2.5}$ concentrations are consistently the highest on the late evening or early morning, peaking between 20 and 40 µg·m$^{-3}$ depending on the day. Concentrations are lowest during the middle afternoon, typically around 6 µg·m$^{-3}$. Such diurnal cycles in aerosol pollution are not uncommon and can be driven by several different mechanisms: (i) diurnal variability in the sources of aerosols (e.g., human activity or photo-oxidation), (ii) variability in transport (i.e., prevailing wind patterns produce effective transport from a point source during certain times of day) or (iii) changes in boundary layer height impacting dilution of relatively constant local emissions. All three mechanisms are important during the more-polluted first half of our measurement period.

First, human activity has a known diurnal cycle that has been observed to impact air quality [46]. An important source of PM$_{2.5}$ in this region is domestic SFU (solid fuel use) for cooking and heating. Most SFU occurs overnight to provide heat during the relatively cold nighttime during winter, as well as for cooking, and this was observed during the measurement period. Black carbon, which is a tracer for combustion, exhibits the same diurnal variability as the total PM$_{2.5}$, suggesting that domestic SFU is the dominant source of combustion-PM$_{2.5}$ in this region. The other major source of black carbon is from diesel vehicles on the A1 highway. However, if this was the dominant source, emissions would peak during times of heavy traffic (the diurnal pattern of traffic in Botswana is not trivial to estimate but likely is lowest overnight). This would not necessarily lead to highest concentrations in the middle of the day, due to differences in boundary layer height and transport; however, the peak in concentrations we observed was well after the vehicle traffic slows down for the night, supporting the hypothesis that SFU is the dominant combustion-aerosol source for this site.

Due to the spatial arrangement of anthropogenic sources of PM$_{2.5}$ relative to our measurement position, wind direction is likely important for PM$_{2.5}$ concentration. In Figure 7, we plot hourly wind measurements from the BIUST campus weather station with temporal subsets as follows: (a) contains the windrose for the entire time period, (b) contains the daytime (7 a.m.–7 p.m.) observations during the first half of the measurement campaign, (c) contains the nighttime measurements (7 p.m.–7 a.m.) from the first half of the campaign, and (d) and (e) contain the daytime and nighttime measurements from the second half of the measurement period, respectively. Throughout the measurement period, the dominant wind direction is from the northeast, and east. Wind speeds are typically higher during the day, and in Figure 7c (first half, nighttime), there are a larger number of hours with southwesterly winds, though most of these measurements are associated with low wind speeds (i.e., stagnant conditions). The wind direction does not appear to follow a diurnal cycle. The closest potentially important point source near the measurement site is the coal power station located to the west-northwest as a source of sulfate for the region. The wind data show that during this observation period, the wind would have rarely transported power plant emissions to our measurement site. Given the relatively slow rate by which SO$_2$ is converted to sulfate in the atmosphere (in the absence of clouds), sulfate is typically a regional-scale pollutant, and does not typically exhibit diurnal variability. Differences in wind behavior do impact sulfate concentrations in this study on longer timescales.

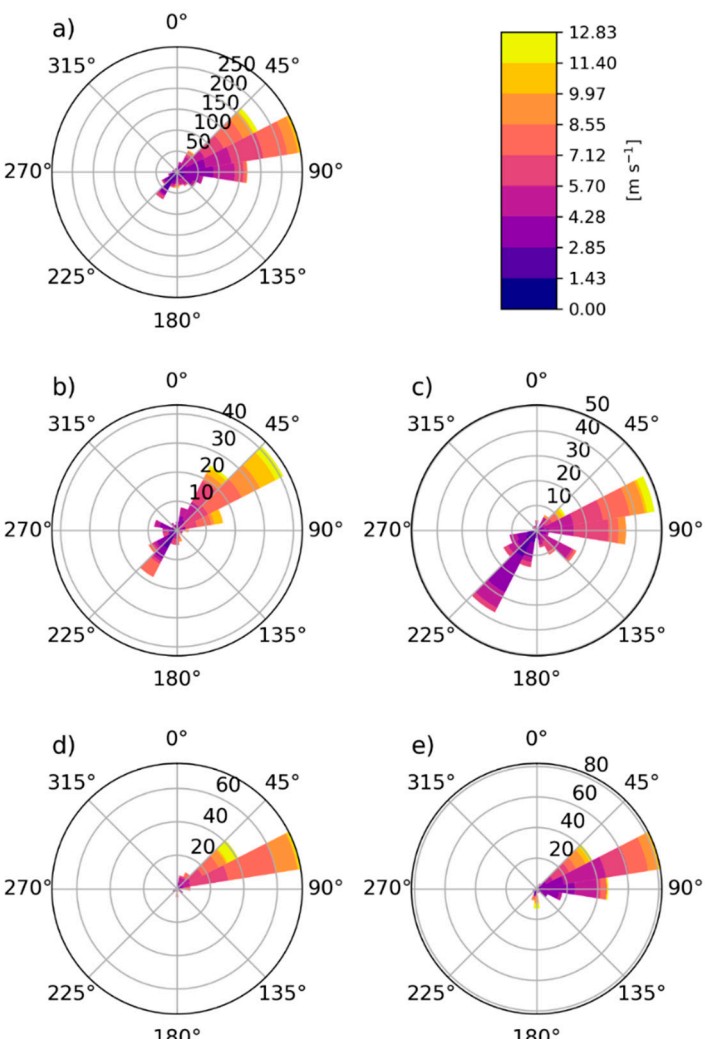

**Figure 7.** Windroses: (**a**) total time period; (**b**) first half, day; (**c**) first half, night; (**d**) second half, day; (**e**) second half, night.

Finally, the diurnal cycle in boundary layer height is likely a key driver in $PM_{2.5}$ and black carbon concentrations at the surface. Due to the warm days with high insolation, the high sensible heat flux from the surface can produce a high boundary layer; likewise, rapid radiative cooling due to the lack of water vapor in the atmosphere can produce cold temperatures and a much shallower boundary layer overnight. The change in boundary layer height impacts the effective volume of air that emissions are diluted into, and for an unchanging emission rate, concentrations increase for a shallow boundary layer. The boundary layer height typically maximizes mid-afternoon when the sensible heat flux is highest, and minimizes at sunrise when the surface radiative cooling is fastest. The maximum and minimum $PM_{2.5}$ concentrations occur early relative to the expected maxima and minima of the boundary layer height, suggesting that there are other factors (such as timing of human behavior and the associated emissions) modulating the effect of boundary layer height. With measurements of boundary layer height, we could constrain the emission flux and timing, which would help to better account for the role local combustion sources are impacting the $PM_{2.5}$ concentration as opposed to advection of regional-scale emissions or upwind point sources. Because boundary layer height depends on surface fluxes as well as larger-scale meteorological conditions, it can be challenging to accurately estimate.

In addition to the diurnal cycle in $PM_{2.5}$ concentrations, there is a big difference in concentrations between the first and second half of the time period. Concentrations are much larger during the first half of the time period (two-week average concentration of 14.0 $\mu g \cdot m^{-3}$) as compared to the second (two-week average concentration 6.5 $\mu g \cdot m^{-3}$). From Figure 3, we see that there is no obvious change in aerosol composition during the two time periods. In Figure 6a, we see that despite the lower concentrations, there is still a diurnal profile associated with black carbon, but the daytime concentrations are close to zero and the overnight concentrations are much lower than during the first half of the measurement period. There was no decrease in overnight temperatures, during the second half of the time period, and consequently there is no reason to believe that there was a systematic change in residential SFU use during this time period. Furthermore, there were no obvious changes in vehicle traffic or on-campus construction activity near the measurement site between the two time periods, though there is no information to thoroughly test this hypothesis. This suggests that the primary driver behind the concentration differences are related to transport patterns and ventilation of emissions away from our measurement location associated with the synoptic or local-scale meteorological conditions. In Figure 7d,e, we show that the second half of the time period was associated with faster wind speeds with more of an easterly contribution; this difference is especially pronounced during night. The differences in wind speed and direction may account for some of the differences between the two time periods; faster winds overnight prevented nighttime concentrations from building up in the second half of the time period, causing a decrease in the 48-h average $PM_{2.5}$ concentration. During the first half of the time period, nighttime winds were occasionally weak and from the southwest; this diurnal wind behavior could help explain the nighttime increases in $PM_{2.5}$ seen during the first half of the time period. Conversely, the second half time period has winds almost uniformly from the east, even at night. East of the measurement site is sparsely populated desert vegetation with no obvious local sources of $PM_{2.5}$.

During the period between 3 July and 8 July during the second half, much of southern Africa experienced cold temperatures and anomalous precipitation due to synoptic weather activity [19]. The cold front passage removed pollution by precipitation and allowed cleaner air to mix into this region. In Figure 8, we plot 24-h HYSPLIT back-trajectories for the two time-periods (Figure 8a shows the first half of the measurement period, Figure 8b shows the second half, note the different spatial scales of the two figures). In Figure 8a, we see that most of the trajectories originate in a very comparatively small spatial area, while in Figure 8b, many of the trajectories originate as far away as South Africa. While the trajectories tend to originate to the east of the measurement site in both cases, the trajectories travel over twice as far in 24 h during the second half. This faster moving air more effectively dilutes emissions. It is possible that during the first half of the measurement, before the frontal passage, a period of regional stagnation allowed local pollution to build up in Palapye. The smaller area of

origin for air masses impacting Palapye during the first half of the time period suggests that regional stagnation may be a factor in the relatively larger concentrations of $PM_{2.5}$ at our measurement site during this time. Therefore, comparisons to remotely sensed products, such as MODIS Aerosol Optical Depth (AOD) could be useful for investigating the importance of stagnation at this location.

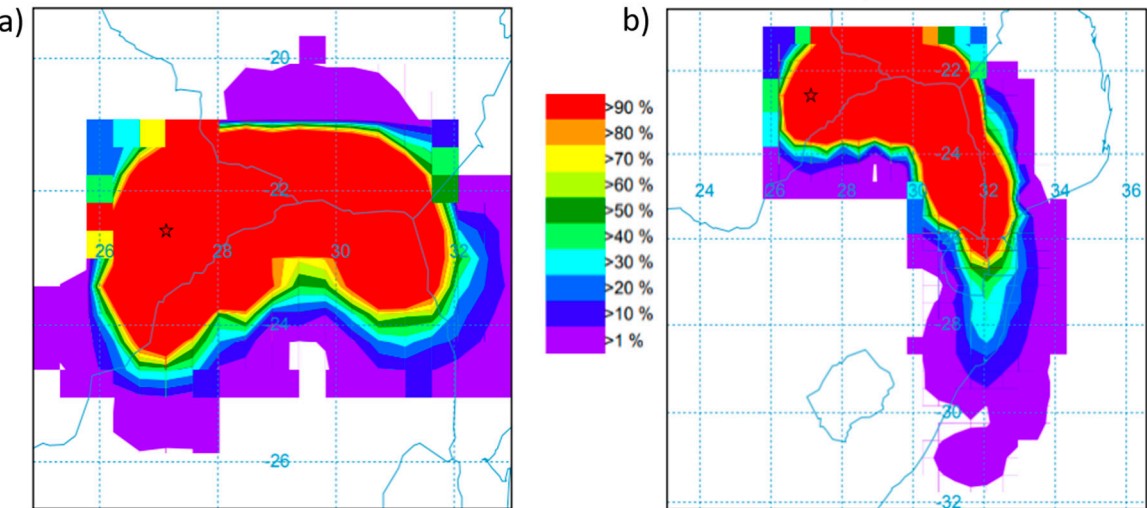

**Figure 8.** HYSPLIT back-trajectories for (**a**) the first half of the measurement period and (**b**) the second half of the measurement period (note that (**b**) spans a much larger area). The star symbol indicates the measurement location.

Finally, we discuss the AOD measurement results. In Figure 9, we show the AMOD AOD observations during our measurement period plotted alongside MODIS observations interpolated to our measurement site, as well as an average of both overpasses over all pixels within 100 km of the measurement site (labelled as "smoothed" in the legend), composited according to the methodology in [47]. First, the AOD observations from all platforms are low relative to more polluted parts of the world, with the highest AODs in this period being less than 0.3. The AMOD AOD is generally higher than MODIS during the entire measurement period. We propose two likely explanations that may explain the discrepancy: first, AMOD measurements were not performed exactly at the Terra or Aqua overpass times, so the difference may be due to temporal offset between the measurement times; second, the MODIS AOD product has not been thoroughly evaluated for this part of the world. The MODIS AOD retrieval can produce inaccurate results over certain surface types, such as highly reflective desert surfaces. The MODIS AOD product uses surface-based AOD measurements, such as the AERONET network to calibrate the AOD retrieval in different locations with varying surface properties, and for different atmospheric conditions. However, there are few AERONET observations in southern Africa; only two sites have data available with a data record longer than two years, and both of these sites are over 600 km away on different land surface types. Therefore, MODIS retrievals for this region may be biased low relative to the true atmospheric loading of aerosols. However, there are also possibilities of errors and bias in the AMOD measurements. While the AMOD has been shown to accurately compare to MODIS and AERONET data in other studies [17,18], there are several reasons these particular AMOD measurements may be biased high, including calibration errors. Therefore, these results should not be interpreted as a robust evaluation of MODIS retrievals for this region, but as an exploratory study and comparison of two separate measurement approaches. Longer term measurements and inter-comparisons to other sun photometers will help to explain the discrepancy between the AMOD and MODIS AOD retrievals. The primary purpose of this field experiment was to establish a long-term data record of surface-based AOD using a different low-cost sun photometer design, data from which will be included in a subsequent study; these data will present a good opportunity to evaluate the AMOD AOD measurements against another low-cost surface-based technique.

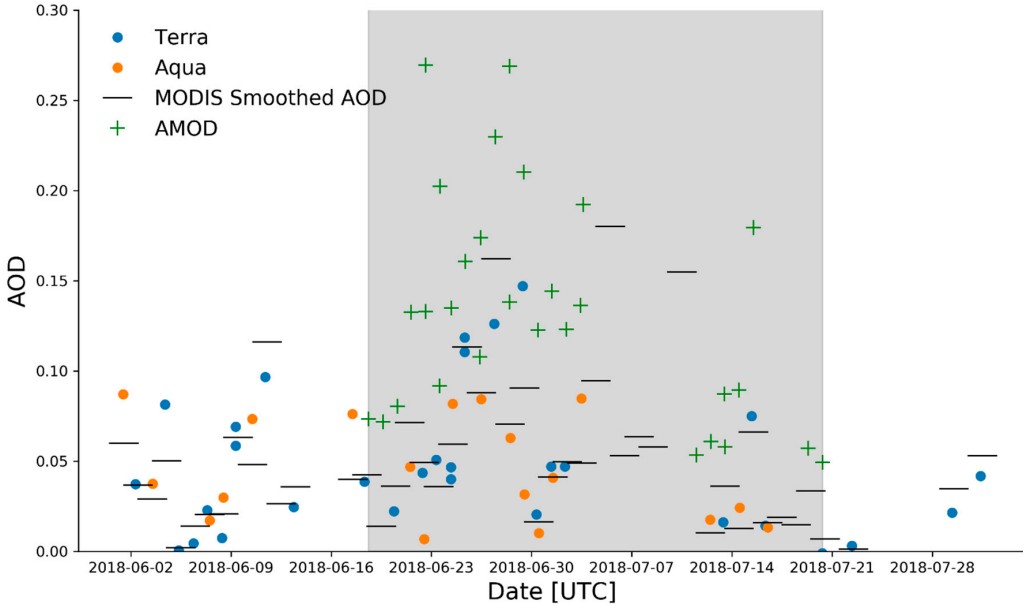

**Figure 9.** MODIS AOD retrieval from Terra (blue) and Aqua (orange) overpasses for Palapye (points) as well as the surrounding 100 km average (black), with AMOD AOD at 520 nm overlaid (green crosses). The shaded gray area refers to the duration of the measurement campaign.

Despite the difference in absolute magnitude for the AOD measurements, the AMOD AOD and MODIS AOD both show the same qualitative result; the AOD is elevated during the first half of the measurement period relative to the second half, which is consistent with the elevated surface concentrations of $PM_{2.5}$ during this period. However, the single-point MODIS observations exhibit a fair amount of day-to-day variability which is unsurprising, given the signal-to-noise ratio inherent to values for AOD around 0.1 over land. In addition to the single-location MODIS AOD, we overlay the average MODIS AOD over a larger spatial area, including all observations in a 50 km radius. In addition to being smoother, the spatial averaging shows a more pronounced increase in AOD during the first half of the measurement period. The spatial averaging helps to reduce the noise in the AOD retrieval, but if the increase in $PM_{2.5}$ were local to Palapye, then it would also wash out the local increases in $PM_{2.5}$. This is not what is observed, as the spatially smoothed MODIS AOD retrieval shows an elevated AOD during the period of elevated surface concentrations. Taken together with the results from the HYSPLIT back-trajectories shown in Figure 8, this suggests that the increase in $PM_{2.5}$ is larger in scale than a local change in emissions near our measurement site in Palapye. Our hypothesis is that this difference between the first and second halves was due to a regional-scale stagnation event during the first half, which was ultimately abated by the passage of a cold front during the first week of July.

Ultimately, it is important to determine the annual average $PM_{2.5}$ at this site, as this is the quantity that is currently most important for estimating the health burden. Given the limited scope of the data we present, there is not a lot that we can conclude about long-term pollution properties, though we may be able to make some informed inferences based on the types of important sources and the climate. Because these measurements were conducted in winter, we experienced cooler temperatures, especially at night (nighttime lows around 0 °C). Enhanced demand for residential heating overnight often leads to higher rates of solid fuel use and associated emissions. Coupled with a cooler and more-stable boundary layer at night, this has led to hazardous $PM_{2.5}$ concentrations in many cities globally. For this reason, we may have been capturing some of the highest concentrations Palapye experiences during a typical year. However, as shown in [34–36], summertime burning in South Africa may be an important regional source for the general region, and remote sensing observations have shown advection of these smoke aerosols to Botswana. However, it is not known if these smoke events impact air quality at

the surface in Palapye. Furthermore, even the period we did experience two different concentration regimes: higher concentrations during regional stagnation, and lower concentrations following the passage of a cold front. Subsequent studies aimed at longer-term monitoring can help to quantify the frequency, duration, and possible seasonality of these stagnation events. Likewise, longer-term studies may determine other local or regional phenomena that we were unable to observe in our brief field campaign.

## 4. Conclusions

In this study, we share results from a serendipitous measurement campaign coinciding with an NSF-IRES field research study at BIUST in Palapye, Botswana. We used low-cost AMOD measurement systems that are capable of collecting $PM_{2.5}$ samples on Teflon filters, along with surface-based AOD, and real-time $PM_{2.5}$ concentrations, all in a portable, low-cost system. In addition, we made $NH_3$ gas-phase measurements using Radiello passive cartridges that were time-integrated over 3.5 days, and a MicroAeth Aethelometer to measure real-time black carbon concentrations. Using these three low-cost instruments for a measurement period of 5 weeks, we were able to make some observations about the concentration and composition of $PM_{2.5}$ on the BIUST campus. According to the more robust filter-based observations, the average $PM_{2.5}$ concentrations during our measurement period were 9.4 $\mu g \cdot m^{-3}$, which is less than the WHO-recommended annual limit of 10 $\mu g \cdot m^{-3}$ and very close to estimate derived from satellite and model data used by the GBD of 9.1 $\mu g \cdot m^{-3}$ [3]. From chemical analysis of the filters, we found that the most important constituent of the particle phase is the inorganic aerosol, which is composed mostly of ammonium sulfate (35% by mass, on average). In addition to inorganic sulfate, a sizeable fraction of the aerosol is composed of carbonaceous species, such as water-soluble organic carbon (12% by mass on average) or black carbon (18% by mass on average). These results are consistent with the number of large coal power plants upwind of the measurement site, as well as with the prevalence of solid fuel combustion in this region. We also determined that dust was not a major component of the $PM_{2.5}$ mixture. Finally, ~35% of the aerosol mass on average was not speciated by any of our analytical techniques. By elimination, we believe that an important fraction of this is likely insoluble organic aerosol; solid-fuel combustion is a source of carbonaceous aerosol, but usually requires time in the atmosphere to age and become soluble in water. Because our extraction of the AMOD filters used water as a solvent, we are unable to test for the presence of insoluble carbonaceous compounds; this is a major shortcoming of our experimental approach.

In addition to the time-integrated filters, we also were able to obtain real-time measurements of $PM_{2.5}$ from the AMOD Planttower sensors and black carbon from the Aethlabs MicroAeth. Both of these measurement principles require assumptions about the optical properties, morphology, and size of the respective particles/substances that are being measured; given these limitations, such measurements provide us with qualitative information about the diurnal variability in BC and $PM_{2.5}$. We observe that the BC and $PM_{2.5}$ are correlated on the diurnal timescale, and that both species exhibit similar diurnal cycles with concentrations maximizing overnight. Based on our observations, our analysis of wind speed and direction measurements from the BIUST campus, and the types of sources that are present, we attribute the diurnal cycle in BC and $PM_{2.5}$ to residential SFU for cooking and heating and changes in boundary layer height likely play a role as well.

Finally, we compared the AMOD AOD measurements with MODIS AOD retrievals. The primary purpose of the study was to establish a longer-term data record of surface-based AOD measurements using a different low-cost design. Those measurements will be shared in the future and are beyond the scope of this 5-week case study. However, they reflect the need for surface-based AOD measurements to help calibrate the MODIS and other satellite-based AOD retrievals while accounting for local differences in surface reflectance. Finally, it is important to accept the limitations of our measurements. The measurements were performed using low-cost instruments, and the data record is short. However, given the lack of available data for this region, we demonstrated how such low-cost measurement campaigns can be used to advance our knowledge on the type, and severity, of aerosol pollution and

sources in a region where no other measurement data are publicly available. We have updated the quality-controlled data and associated metadata to a publicly available data repository [48].

**Author Contributions:** Conceptualization, J.R.P., G.M.T., J.P.S. and S.B.; Formal analysis, W.L., J.P., E.J.B., A.P.S., B.F., J.P.S. and S.B.; Funding acquisition, G.M.T., S.B., J.P.S.; Investigation, W.L., J.P.S., J.L.C.J. and S.B.; Methodology, W.L., J.R.P., B.F. and J.P.S.; Project administration, J.R.P., J.P.S. and S.B.; Resources, G.M.T., J.P.S., J.L.C.J. and S.B.; Supervision, J.R.P., G.M.T. and J.L.C.J.; Writing—original draft, W.L.; Writing—review & editing, J.R.P., A.P.S., G.M.T., J.P.S., J.L.C.J., B.F. and S.B. All authors have read and agreed to the published version of the manuscript.

**Funding:** We would like to acknowledge the NSF-IRES Grant number 1559308 for providing the financial support for the travel and logistics for conducting these measurements. We would like to acknowledge NASA grant nos. NNX17AF94A and 80NSSC18M0120 for supporting the development of the AMOD.

**Acknowledgments:** We would like to acknowledge our colleagues and hosts at BIUST for their scientific support and hospitality. We would especially like to mention Elisha Shemang for his assistance with the visit logistics, and Serowalo Mokgosi for providing laboratory space for our research operations. We would like to acknowledge the BIUST students who participated in the sun photometer measurements including: Samuel Lebengwa, Hope Solomon, Bosa Mosekiemang, Fekadu Demisse, Mulugeta Melaku, Anteneh Getachew, and many others. We would like to acknowledge the other students who travelled for the project: Ian Krintz (Appalachian State University), Brandon Lewis (North Carolina State University), Julian Gordon (North Carolina A&T University), and Ana Carrell (North Carolina A&T University).

**Conflicts of Interest:** The authors declare no conflict of interest.

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
