# Peer review of "Using Low-Cost Measurement Systems to Investigate Air Quality: A Case Study in Palapye, Botswana"

_atmosphere, doi:10.3390/atmos11060583_

Round 1
Reviewer 1 Report
The study seems interesting and needed. The potential for low-cost instruments in air quality monitoring is growing, especially in developing countries. Although the article is appropriately written, it demands some improvements to make it more clear for the readers.
The list of abbreviations is very needed; there are too many abbreviations in the text to remember them during reading.
The fragment 235-237 should be explained in detail.
Specific comments:
References need to be corrected [17] and [18] are not in the order they are in line 193, while [19] is in line 129.
There are names as reference in lines: 131, 133, 236, 255, 500, 542
The space is missing 262, 263, 264, 485, 541, 542
There is double space: 168,204, 205, 206, 208,209, 227, 336, 412
Line 102 )
Line 172 tthe
The units are not uniform µg/m3 or µg m-3, mL or ml
Line 210, please explain why 1.6.
Line 290 ages?
Figures 3 and 4 why the bars overlap, the y-axis needs to add which filter/aerosol is presented
Figures 4 and 7 are before the text,
Fragment 282 (ii) needs a reference
Fragment 303-305 needs a reference
Line 342 needs explanation
Fragment 350-352 is not clear
Line 360 metallic species only?
Author Response
Please see attachment, section "Reviewer 1".
Responses are in red.

Reviewer 2 Report
My review is attached as a pdf file.

Author Response
Please see attachment, section "Reviewer 2".
Responses are in red.

Reviewer 3 Report
This manuscript investigates the air pollution levels using low cost sensor measurements in Palapye, Botswana. The manuscript is well written but have many grammatically and typos error. Overall, the manuscript is relevance and interest to the Atmosphere readers. I recommend this paper for publication after minor corrections (listed).
1. Abstract- It should be defiantly improved. Please summarize results, not only measurements.
2. Accuracy, validity and error correction of low cost sensors- Please provide these information in the manuscript.
3. Figure 8- It needs more explanation in the text as it includes key results.
Author Response
Please see attachment, section "Reviewer 3".
Responses are in red.
